# The rs2228145 Variant of the Interleukin-6 Receptor (IL-6R) Gene Impacts on In Vitro Cellular Responses to SARS-CoV-2 VOC B1.1.7 Recombinant Spike Protein

Saira Sarwar, Rebecca Aicheler , Lee Butcher , Katie Rees, Stephen Potter, Richard Rowlands and Richard Webb *

Department of Biomedical Sciences, School of Sport and Health Sciences, Cardiff Metropolitan University, Cardiff CF5 2YB, UK; ssarwar2@cardiffmet.ac.uk (S.S.); raicheler@cardiffmet.ac.uk (R.A.); lbutcher@cardiffmet.ac.uk (L.B.); krees@cardiffmet.ac.uk (K.R.); spotter@cardiffmet.ac.uk (S.P.); rrowlands@cardiffmet.ac.uk (R.R.)
* Correspondence: rwebb@cardiffmet.ac.uk; Tel.: +44-2920-205559

**Abstract:** Given the variability in inflammatory responses to SARS-CoV-2 infection observed within human populations, we aimed to develop an in vitro model system (based on monocyte-macrophages, a key relevant cell type) that could yield insights regarding the impact of rs2228145, a clinically relevant polymorphism within the coding region of a key inflammatory gene in the body's response to SARS-CoV-2 infection: the interleukin-6 receptor (IL-6R) gene. Three monocyte-macrophage cell-lines (U937, THP-1, MM6) were shown to exhibit AA, AC and CC rs2228145 genotypes, respectively, and to exhibit an MM6 > THP-1 > U937 pattern regarding basal levels of soluble IL-6R (sIL-6R) release. Similar MM6 > THP-1 > U937 patterns were seen regarding the extents to which (i) circulating levels of the IL-6/sIL-6R 'active complex' increased and (ii) phosphorylation of the downstream transcription-factor STAT3 occurred, following treatment with SARS-CoV-2 spike protein (SP). Moreover, a blocking antibody for the ACE-2 entry receptor for SARS-CoV-2 suppressed effects (i) and (ii), suggesting that interaction between SP and ACE-2 is the initial event that triggers IL-6/IL-6R signalling in our system. Production of IL-8 occurred to greater extents in A549 lung epithelial cells treated with tissue-culture supernatants from SP-treated MM6 cultures than SP-treated THP-1 or U937 cultures. Our data indicate that the rs2228145 genotype significantly impacts upon SP-associated IL-6/sIL-6R signalling in vitro, suggesting that it may influence in vivo risk of developing severe COVID-19 and/or long-COVID symptoms following infection by SARS-CoV-2. Thus, the rs2228145 genotype may have potential as a biomarker that differentiates between patients at risk of developing severe and/or prolonged symptoms following infection by SARS-CoV-2 and those who are at less risk.

**Keywords:** IL-6; sIL-6R; COVID-19; long-COVID; monocyte-macrophage; genetic variants

## 1. Introduction

Responses to COVID-19 are variable: most SARS-CoV-2-infected individuals have mild or no symptoms, but in ~10–15% of cases COVID-19 patients progress to severe disease, in ~5% to critical disease and in ~1.3% to lethality [1,2]. Lethality rises with age and the pre-existence of co-morbidities [1,3], but these two factors do not explain all of the observed variability, with variation in inflammatory responses being a key additional factor [1]. Therefore, evaluation of the impact of variability in inflammation-associated genes on responses to SARS-CoV-2 infection is a key priority.

The immune system is essential for clearance of SARS-CoV-2 from the infected host and thus resolution of clinical symptoms. However, autopsies of deceased COVID-19 patients have revealed profound inflammation but negligible levels of SARS-CoV-2 in lungs, vasculature, brain and other organs, suggesting that these patients died with pathologic hyper-immune activation despite effective viral clearance [4]. Thus, activated immune cells

and excessive inflammatory responses are key players in COVID-related mortality [5]. In particular, monocyte-macrophages have been implicated in the excessive COVID-19 inflammatory responses known as 'COVID-19 cytokine-storms' (COVID-CSs) [1,4,6]. COVID-CSs are a major cause of disease severity and death in patients with COVID-19; their occurrence is associated with severe consequences such as acute respiratory distress syndrome, multi-organ failure and even death [6–8]. Accordingly, a major clinical priority is to identify how (and when) COVID-CSs can be prevented, interrupted or suppressed [8].

It should be noted that migration of virus-containing monocyte-macrophages from the lungs is thought to be key to the spreading of COVID-19 to other organs [6]. For this reason, McGonagle has proposed the term 'Macrophage Activation Syndrome' be applied to hyper-inflammatory COVID-19 responses, while 'immunopathology' is regarded as a key contributor to development of both severe COVID-19-related pulmonary disease/systemic symptoms [7,8] and post-COVID-19 condition (so-called 'long-COVID'; defined by the WHO as persistence, for at least 12 weeks from onset of COVID-19, of symptoms that cannot be explained by an alternative diagnosis) [9–11].

Interleukin-6 (IL-6) is an important inflammatory cytokine that has been associated with severe cases of COVID-19 [12]. IL-6 can signal via two forms of its receptor: full-length IL-6R found bound within cell membranes (mbIL-6R) leading to classical IL-6 signalling and a truncated version shed from cells into the circulation as soluble IL-6R (sIL-6R) resulting in trans IL-6 signalling [13]. Such shedding is catalysed by proteolytic enzymes, primarily ADAM-17 [14]. Evidence is emerging that circulating levels of IL-6 and/or sIL-6R correlate with the severity of COVID-19 symptoms and with clinical outcomes attributable to excessive COVID-19 inflammatory responses, and hence to risk of COVID-CSs [15,16]. For example, patients requiring ICU admission have significantly higher levels of serum IL-6 and sIL-6R levels [1,6,16], while Tocilizumab (a drug that can block both forms of IL-6 signalling) has been used as an effective treatment for severe COVID-19 [17–20].

According to the National Center for Biotechnology Information, the IL-6R gene has a single clinically-relevant single nucleotide polymorphism (SNP) in its coding region: this variant (designated rs2228145) involves a genetic change from A to C at position 1073 of the gene [21]. This change at the nucleotide level is linked to an $Asp^{358} \rightarrow Ala^{358}$ change in the IL-6R protein, which enhances cleavage of mbIL-6R at the adjacent cleavage site ($Val^{356}$), and hence increases sIL-6R shedding [21,22]. Population genetics studies have reported that ~40% of the global population bear the rs2228145 AA genotype, ~45% the AC genotype and ~15% the CC genotype (with some variation being present in different ethnic groups) [22]. Accordingly, previous studies have argued that the rs2228145 genotype may exacerbate inflammatory disease risk for large numbers of patients by impacting on their circulating sIL-6R levels [23,24]. Similarly, the rs2228145 genotype may influence the risk of developing severe COVID-19 symptoms on infection by SARS-CoV-2; for this reason, Chen et al. have stated that examination of the rs2228145 polymorphism in the context of COVID-19 is warranted [4]. Using the rs2228145 genotype as a predictive biomarker of COVID-19 severity may be of use in helping differentiate between patients at risk of developing severe COVID-19 symptoms upon infection by SARS-CoV-2 and those who are at less risk, and so may bring advantages regarding management and treatment of COVID-19 patients.

While the successful vaccine roll-out in many countries has arguably reduced the need for emergency measures regarding treatment of COVID-CSs [25], post-COVID-19 condition or 'long-COVID' provides an additional context where there is an urgent need to improve the effectiveness of management/treatment. Long-COVID's global prevalence has been estimated to be approximately 200 million affected individuals worldwide [2]. Clearly, this constitutes a major societal problem, and measures for identifying individuals who are likely to be susceptible to long-COVID are urgently needed. Individuals' cytokine responses to SARS-CoV-2 may influence the likelihood of developing not only acute COVID-19 but also long-COVID: for example, given that IL-6 and sIL-6R are implicated in neuroinflammatory processes in a number of contexts [26–29], neuropsychiatric

long-COVID symptoms may be attributable at least in part to IL-6/sIL-6R signalling [10,11]. This has recently been supported by the REMAP-CAP Long-Term Follow-Up study [30,31], which reported that Tocilizumab significantly improved several aspects of health-related quality of life (including cognitive function) six months after initial SARS-CoV-2 infection, while Tocilizumab was protective against neuropsychiatric symptoms in a cohort of COVID-19 survivors three months after discharge following hospitalisation [32]. This link to IL-6/IL-6R signalling further supports the potential use of the rs2228145 genotype as a predictive biomarker to identify at-risk patients (at risk of long-COVID, in this case), for whom IL-6 signalling blockers such as Tocilizumab could be appropriate therapies.

Therefore, we investigated whether the rs2228145 genotype impacted on cellular responses to SARS-CoV-2. We focused on monocyte-macrophages, which as described above are a key cell type in the pathology of COVID-19: they are major sources of IL-6 and sIL-6R [6]; they express the SARS-CoV-2 entry receptor, ACE2 [6]; and they are susceptible to infection by SARS-CoV-2, with infected cells expressing increased levels of IL-6 [6]. It should also be noted, given the importance of long-COVID's neuropsychiatric symptoms (see above), that monocyte-macrophages can infiltrate the CNS, take on microglial phenotypes and participate in neuroinflammatory processes [33,34]. Treatment of monocyte-macrophages with recombinant SARS-CoV-2 spike protein (SP) was used as an in vitro mimic of SARS-CoV-2 infection. This was originally acknowledged, and utilised, as a valid experimental approach following the demonstration that SARS-CoV SP both bound to ACE2 on the monocyte-macrophage cell surface [23] and significantly increased monocyte-macrophage IL-6 mRNA expression and secretion via activation of the ACE2/AT-1/NFkB signal transduction pathway [24]. As expected, given the SARS-CoV-2 SP's ability to bind ACE2 with higher affinity and faster binding kinetics than that of SARS-CoV [25], this approach has also proved useful in the investigation of SARS-CoV-2. Shirato (2021) reported NFkB-mediated IL-6 signalling in SARS-CoV-2-treated macrophages [12], while Patra (2020), and Karwaciak (2021) reported comparable responses to viral infection and SP treatment [14,35]. Specifically, comparable NFkB-mediated increases were observed for IL-6 release, ADAM-17 expression and sIL-6R shedding, which secondarily induced STAT3 phosphorylation and downstream cytokine release in other cell types (such as endothelial cells [14] and Th17 lymphocytes [35]). Therefore, study of this ACE2-NFkB-IL-6/sIL-6R-STAT3 pathway was a priority for the present study.

By showing that three distinct monocyte-macrophage cell lines (THP-1, MM6, U937 [36–38]) exhibit the AA, AC and CC rs2228145 genotypes, respectively, we have developed an in vitro model system for investigating the impact of the IL-6R variant rs2228145 on monocyte-macrophage function, in the context of COVID-19 pathology. We aimed to utilise this model system to test our hypothesis that the rs2228145 genotype influences the likelihood of excessive COVID-19 inflammatory responses by evaluating the respective impacts of the SARS-CoV-2 SP on THP-1, MM6 and U937 cells. Firstly, we investigated SP-induced signal transduction mechanisms (as described above, these are reported to involve the ACE2-AT1-NFkB pathway [12,14,35]). Secondly, we evaluated the IL-6 signalling-related consequences of these signal transduction processes in THP-1, MM6 and U937 cultures—namely, release of IL-6, sIL-6R and the associated signalling protein sgp130. Finally, we focused on downstream responses to such SP-evoked IL-6 signalling effects. In the monocyte-macrophages themselves, we assessed phosphorylation of STAT3 as a downstream target of IL-6 signalling, while via conditioned-media experiments, we elucidated the IL-8 responses of A549 lung epithelial cultures to tissue-culture supernatants collected from the three monocyte-macrophage cell lines ± SP treatment.

## 2. Materials and Methods

### 2.1. Materials

All reagents were purchased from Sigma-Aldrich (Poole, UK), unless stated otherwise. HEK293-derived His-tagged recombinant SARS-CoV-2 spike protein (B.1.1.7 (accession #: YP_009724390.1)) was obtained from R&D Systems (Cambridge, UK; please note that this

B.1.1.7 variant is a variant of concern distinct from the original Wuhan strain and that responses to this variant may differ from responses to the original strain). THP-1, MM6 and U937 cells were obtained from the European Collection of Authenticated Cell Cultures (ECACC). IL-6, sIL-6R, sgp130 and IL-8 ELISA DuoSets (#DY206, #DY227, #DY228 and #DY208) were purchased from R&D Systems (Cambridge, UK). AF933 ACE-2 blocking antibody was purchased from R&D Systems (Cambridge, UK). The following antibodies were purchased for use in Western blot experiments: EPR17622 pNFkB Ser 276 (ab183559) EP2161Y NFkB (ab76311) (both from Abcam (Cambridge, UK); D3A7 pSTAT3 Tyr 705 (#9145S); D1A5 STAT3 (#8768); D6A8 Beta-actin (#8457T); HRP-linked anti-rabbit IgG secondary antibody (#7074) (all from Cell Signalling (Leiden, Netherlands)). The following fluorophore-conjugated antibodies for use in flow cytometry experiments were purchased from Biotechne (Oxford, UK): fluorescently conjugated antibodies to ACE2 (FAB9332G anti-human ACE-2 Alexafluor conjugated antibody, plus IC003G mouse IgG2a Alexafluor isotype control) and ADAM-17 (MAB9301 anti-human TACE2/ADAM-17 ectodomain primary antibody/MAB002 mouse IgG2a isotype control, plus F0101B APC-conjugated anti-human IgG secondary antibody).

### 2.2. Maintenance, Differentiation and Treatment of Cell Lines

Standard protocols [39] were followed to culture MM6, THP-1 and U937 cell lines in RPMI media. As described previously [40], monocytic cells from the three cell lines were treated for 72 h with 50 ng/mL PMA to ensure complete differentiation to macrophage-like cells (referred to as dMM6, dTHP-1 and dU937). Cells were treated for 24 h with recombinant SARS-CoV-2 SP. As Barhoumi (2021) reported that SP doses of 100 nM led to significantly increased levels of apoptosis [41], whereas Buzhdygan (2020) reported no adverse effect on cell viability with 10 nM [42], we used doses of 0–10 nM SP in our experiments; this did not induce any decreases in cell viability. In some experiments, LPS (24 h; 10 ng/mL) was used as a positive control for eliciting NFkB activation, IL-6 and/or sIL-6R release or STAT3 phosphorylation. For conditioned media experiments, A549 lung epithelial cells were cultured in DMEM media. A549 cultures were treated (24 h) with supernatants from U937, THP-1 and MM6 monocytic cultures that had been exposed to SP (0–10 nM; 24 h) or were treated directly with recombinant SP (10 nM) or recombinant IL-6 and sIL-6R (50 ng/mL in both case).

### 2.3. Genotyping Assay

IL-6R rs2228145 genotype was determined as described previously [43]. Primers flanking the rs2228145 site were designed using Primer-BLAST software (https://www.ncbi.nlm.nih.gov/primer-blast; accessed 22 July 2023): Forward: 5′-TGACAGCACCAGCTAAGT-3′; Reverse: 5′-ACAATGGCAATGCAGAGGAG-3′. Primers plus 1.78 µg DNA template were included within 25 µL PCR reactions (95 °C/3 min; 35 cycles of 95 °C/30 s, 61 °C/30 s, 72 °C/60 s; 72 °C/5 min). PCR products were subjected to HinfI restriction digests (5 µL CutSmart buffer (10×); 1 µL HinfI; 1 µg DNA, within 50 µL reaction volumes; 37 °C; 60 min). Given the presence of the HinfI consensus sequence (GATTC) in PCR product from AA homozygotes, but its absence in CC homozygotes (GCTTC), samples' banding patterns in agarose gels reflected the genotype of the sample donor: either an intact PCR product (183 bp (CC homozygotes)), a pair of fragments (101 bp and 82 bp (AA homozygotes)) or a triplet of bands (183 bp, 101 bp, and 82 bp (AC heterozygotes)).

### 2.4. Western Blot

Western blot experiments were carried out as described previously [44]. Briefly, total protein extracts from THP-1, U937 or MM6 cells (treated for 24 h with 0–10 nM SP $\pm$ pre-treatment for 2 h with 2–20 µg/mL ACE2 blocking antibody) were prepared using 100 µL of protein extraction/lysis buffer, containing 1 mM protease inhibitor cocktail and 1 mg/mL phosphatase inhibitor (Active Motif Ltd., Rixensart, Belgium). Protein content was then estimated using a BCA protein assay (Bio-Rad Laboratories, Basingstoke, UK). Samples

(30 µg protein in each case) were subjected to SDS-PAGE, transferred to nitrocellulose membranes and probed with primary antibodies (16 h; see below for dilutions), followed by HRP-labelled anti-rabbit IgG antibody (2 h; 1:2000 dilution; Cell Signalling Tech., Danvers, MA, USA). Immunogenic bands were detected via enhanced chemiluminescence (West Pico Dura Luminol/enhancer substrate, Rockford, IL, USA) using a Fusion FX system (Collegien, France). The following primary antibodies were used (at 1:1000 dilutions in all cases): anti-phospho-STAT3 $Y^{705}$, anti-STAT3, anti-phospho-NFkB p65 $S^{276}$, anti-NFkB p65. Anti-β-actin antibodies (1:1000; Cell Signalling Tech., Danvers, MA, USA) were used for normalisation purposes, in order to confirm equal loading of samples. Stripping and re-probing of gels using Restore Stripping Buffer (Fisher Scientific, Loughborough, UK) allowed visualisation and comparison of the intensities of different immunogenic bands on each gel. Activation of STAT3 was expressed as the ratio of phosphorylated to total STAT3, as detected as a doublet of immunogenic bands at ~90 KDa and a single immunogenic band at ~80 KDa, respectively. Similarly, activation of NFkB was expressed as the ratio of phosphorylated to total NFkB p65, as detected as immunogenic bands at ~80 KDa and ~65 KDa. B-actin was detected as an ~45 KDa immunogenic band. Quantitative densitometric band-intensity data was obtained using Image J v1.8.0_172 software.

### 2.5. Flow Cytometry

Fluorescently conjugated antibodies to ACE2 (FAB9332G anti-human ACE-2 Alexafluor$_{488}$-conjugated antibody) or IC003G mouse IgG2a Alexafluor$_{488}$ isotype control were used to determine surface expression levels of ACE2 in dU937, dTHP-1 and dMM6. Samples (labelled as 'cells only', 'isotype control' and 'ACE-2'; 10,000 events per sample) were analysed using a Beckman Coulter CytoFlex flow cytometer, using the following fluorescence specifications: Alexa Fluor: $\lambda_{Ex}$: 488 nm, $\lambda_{Em}$: 520 nm. An indirect approach involving MAB9301 TACE/ADAM-17 ectodomain antibody plus F0101B APC-conjugated secondary antibody was used to determine surface expression levels of ADAM-17 in dU937, dTHP-1 and dMM6, using the following fluorescence specifications: APC: $\lambda_{Ex}$: 594/633 nm, $\lambda_{Em}$: 660 nm.

### 2.6. Bioinformatics

Sequence analyses and alignments to identify NFkB-REs (5′-GGGATTTTCC-3′) in promotor regions of IL-6 and ADAM-17 genes were performed using DNASTAR software (Lasergene, version 7; DNASTAR Inc., Madison, WI, USA). All sequences were obtained from National Center for Biotechnology Information databases (http://www.ncbi.nlm.nih.gov/; accessed 22 July 2023).

### 2.7. DuoSet ELISA: Human IL-6, Human sIL-6R, Human sgp130 and Human IL-8

After treatment with SP (0–10 nM; 24 h ± pre-treatment for 2 h with 2–20 µg/mL ACE2 blocking antibody), the concentrations of IL-6, sIL-6R or sgp130 secreted by THP-1, U937 and MM6 or dTHP-1, dU937 and dMM6 cells were measured using ELISA, using commercially available duoset kits, as per the manufacturer's instructions. For conditioned media experiments, IL-8 released from A549 cultures treated with supernatants from U937, THP-1 and MM6 monocytic cultures that had been exposed to SP (0–10 nM; 24 h), or were treated directly for 24 h with 10 nM recombinant SP or 50 ng/mL recombinant IL-6/sIL-6R, was determined using ELISA, using commercially available IL-8 duoset kits, as per the manufacturer's instructions. A post-hoc calculation, namely subtraction of the IL-8 concentration seen in the U937, THP-1 or MM6 supernatant (which contains IL-8 released from the monocytic cells) from that seen in the A549 supernatant (which contains IL-8 released from both cell-types), was used in each case to determine the IL-8 release from A549 cells.

To determine the levels of IL-6 complexed with sIL-6R (i.e., the concentrations of the IL-6/sIL-6R 'active complex'), we used the formula described previously [45]:

$$[\text{IL-6} \bullet \text{sIL-6R}] = 0.5[\text{sIL-6R}]_i + 0.5[\text{IL-6}]_i + 0.5K_D - 0.5([\text{sIL-6R}]_i^2 + [\text{IL-6}]_i^2 + 2[\text{IL-6}]_i K_D + K_D^2)^{0.5}$$

where $[\text{sIL-6R}]_i$ and $[\text{IL-6}]_i$ represent the initially used concentrations, and $K_D$ for the formation of IL-6•sIL-6R was taken to be 500 pM, as reported previously [45].

*2.8. Statistical Analysis*

Data are expressed as mean ± standard deviation, unless stated otherwise. Minitab 19.1 (2019) software was used to statistically analyse the data obtained, with confidence levels set at 95% (significance level $p \leq 0.05$). Two-sample *t*-tests and one-way ANOVA analyses with post-hoc Dunnett's comparisons (or Kruskal–Wallis tests for non-parametric analyses) were conducted to compare two-sample and multiple-sample datasets, respectively. Two-way ANOVAs were also conducted to determine whether multiple parameters exerted significant influences on the experimental factor under investigation; such analyses typically involved (i) cell-type (and hence rs2228145 genotype) and (ii) SP dose.

## 3. Results

As shown in Figure 1, agarose gel banding patterns revealed that the three cell lines expressed the following rs2228145 genotypes—U937: AA, THP-1: AC, MM6: CC. Because they span the range of genotypes that occur in vivo, we were therefore able to use SP-treated samples of these three cell lines as an in vitro model system to gain insights regarding the influence of the rs2228145 genotype on the responses of monocyte-macrophages from patients who have been exposed to SARS-CoV-2.

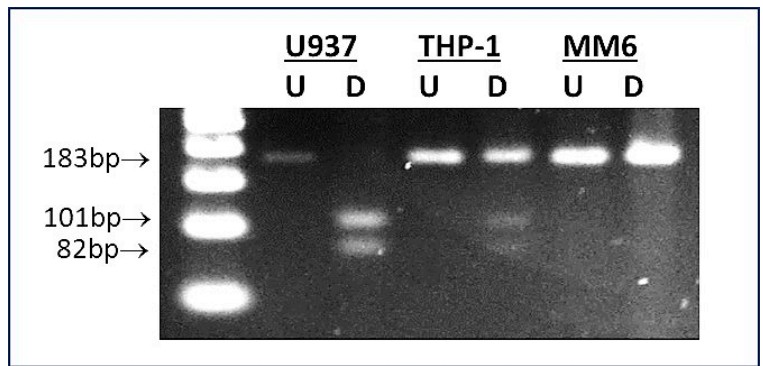

**Figure 1. Determination of rs2228145 genotype of U937, THP-1 and MM6 monocyte-macrophage cell lines**. A representative image is shown of an agarose gel containing U937, THP-1 and MM6 samples subjected to PCR/RFLP-based rs2228145 genotyping assay [43]. Lane 1: 500 bp DNA ladder; lanes 2 and 3: U937 (undigested (U) and digested (D), showing banding pattern indicative of AA genotype (i.e., PCR product cleaved into two fragments: 101 bp and 82 bp)); lanes 4 and 5: THP-1 (undigested (U) and digested (D), showing banding pattern indicative of AC genotype (i.e., intact PCR product plus two cleavage products, therefore three bands: 183 bp, 101 bp and 82 bp)); lanes 6 and 7: MM6 (undigested (U) and digested (D), showing banding pattern indicative of CC genotype (i.e., intact PCR product: 183 bp)).

Flow cytometry was used to gain insights regarding ACE2 and ADAM-17 expression levels within the three cell lines. As shown in Supplementary Figure S1A–C, ACE2 expression was detected in all three cases, with no significant differences in expression levels observed between dU937, dTHP-1 and dMM6 (untreated samples in all cases; $p > 0.05$; Kruskal–Wallis). This indicates that the three cell lines express comparable levels of ACE2 on their surfaces and suggests that they would be expected to bind to SP to approximately similar extents. Similarly, flow cytometry indicated that surface expression of ADAM-17 was evident at comparable levels in dU937, dTHP-1 and dMM6 (Supplementary Figure S1D–F; untreated samples in all cases; $p > 0.05$; Kruskal–Wallis).

It has previously been reported that interaction between SARS-CoV-2 SP and ACE2 initiates signalling responses via AT1 and NFkB [12,14,35]. We investigated phosphorylation of NFkB p65's Ser276 residue using Western blotting and found that, while the three cell lines showed differing basal levels of NFkB p65 phosphorylation, 24 h treatment with 10 ng/mL LPS or 1 nM SP appeared to be linked to increases in phosphorylation (albeit with this effect only achieving statistical significance for LPS ($p < 0.05$; Supplementary Figure S2)). Our bioinformatics analysis identified NFkB response element sequences (NFkB-REs) at positions $-136$ and $-4285$ of the IL-6 gene and at position $-1073$ of the ADAM-17 gene (inset, Supplementary Figure S2), suggesting that both IL-6 and ADAM-17 are encoded by 'target genes' for NFkB signalling.

Given their similar expression levels of ADAM-17 (the prime 'sheddase' responsible for cleavage of mbIL-6R [14]), the three cell lines might be expected to release sIL-6R to similar extents. However, previous studies have reported unchanged expression of full-length IL-6R mRNA in primary PBMCs from healthy volunteers (whose rs2228145 genotypes were AA (33%)/AC (50%)/CC (17%)) [24], but significant genotype-dependent differences in circulating sIL-6R levels, [13,21,22,24,43]. This implies that rs2228145 impacts on full-length mbIL-6R's sensitivity to proteolytic cleavage, and hence on sIL-6R shedding. Therefore, in the current study, we hypothesised that the three cell lines' different rs2228145 genotypes would be associated with varying levels of sIL-6R shedding, and indeed, a statistically significant MM6 > THP-1 > U937 pattern in basal sIL-6R levels was evident in tissue-culture supernatants from untreated cell line cultures (Figure 2A; $p < 0.05$, one-way ANOVA).

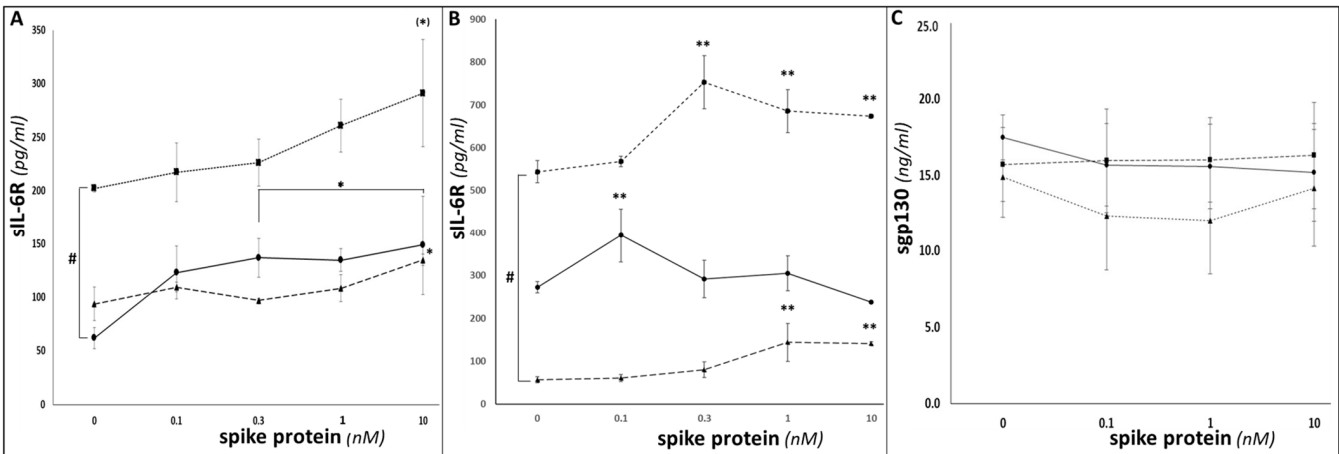

**Figure 2. Determination of sIL-6R shedding in U937, THP-1 and MM6.** ELISA duoset assays were conducted as per manufacturers' instructions to determine sIL-6R levels in supernatants from U937 (circles; solid line), THP-1 (triangles; dashed line) and MM6 (squares; dotted line) monocytic cultures (**A**) or differentiated dU937 (circles; solid line), dTHP-1 (triangles; dotted line) and dMM6 (squares; dashed line) macrophage cultures (**B**), following treatment for 24 h with SP. (**C**) illustrates data from ELISA duoset assays to determine the impact of 24 h SP treatment on levels of sgp130 in supernatants from U937 (circles; solid line), THP-1 (triangles; dashed line) and MM6 (squares; dotted line) cultures. ($n = 6$ in all cases; ** denotes $p < 0.01$; * or # denotes $p < 0.05$; (*) denotes $p = 0.08$; asterisks denote comparisons versus the untreated control for the cell line in question; hash symbols denote comparisons between cell-lines).

SP-elicited dose-dependent sIL-6R increases were seen; given the presence of an NFkB-RE in the ADAM-17 promoter region, these may reflect NFkB-mediated ADAM-17 upregulation and consequent increased ADAM-17-catalysed proteolytic cleavage of IL-6R. These increases either achieved or approached statistical significance ($p < 0.05$ for U937 and THP-1; $p = 0.08$ for MM6, one-way ANOVA in all cases) for all three monocytic cell lines. Importantly, the above MM6 > THP-1 > U937 pattern was maintained, with MM6 showing greater levels of sIL-6R compared to the other two cell lines throughout the range of SP

doses investigated. Thus, both SP dose and cell-type/rs2228145 genotype significantly influenced how much sIL-6R was shed from the cell surface ($p < 0.001$, two-way ANOVA). Significant increases in sIL-6R above control levels were also seen in SP-treated macrophage samples ($p < 0.05$, one-way ANOVA; Figure 2B). SP dose and rs2228145 genotype/cell-type interacted to significantly influence macrophage sIL-6R responses to SP treatment, with dMM6 exhibiting significantly stronger responses than either dU937 or dTHP-1 ($p < 0.05$, two-way ANOVA).

IL-6/sIL-6R trans-signalling can be inhibited by binding to sgp130, with the resulting IL-6/sIL-6R/sgp130 complex being referred to as a 'buffer complex' or 'inactive complex' [45–47]. Therefore, we measured sgp130 levels in tissue-culture supernatants from SP-treated U937, THP-1 and MM6 samples. Unlike for sIL-6R (or for IL-6—see below), no significant differences in sgp130 levels were seen between the different cell types or between the presence or absence of SP treatment, with values of 10–20 ng/mL being seen in all samples ($p > 0.05$ for both genotype and SP dose, two-way ANOVA; Figure 2C).

Increases in IL-6 release were also seen in response to SP treatment (Figure 3A); as discussed above, this may be tentatively linked to SP-triggered/NFkB-mediated increases in transcription of the IL-6 gene [8,10,35], given the presence of NFkB-REs in the IL-6 promoter region. For monocytic cells, MM6 IL-6 release increased in a significant dose-dependent manner ($p < 0.001$), and THP-1 showed a lesser, but still statistically significant, increase ($p < 0.01$), whereas U937 cells did not show a significant increase. (Similarly, treatment with LPS (10 ng/mL; 24 h) as a positive control elicited significant IL-6 increases for THP-1 and MM6, but not U937). Across the whole dataset, two-way ANOVA analysis showed that both SP dose and cell type exerted significant influences on IL-6 release ($p < 0.001$). Dose-dependent statistically significant increases in IL-6 release from differentiated dU937, dTHP-1 and dMM6 macrophage-like cells were seen following SP treatment in all three differentiated cell lines ($p < 0.05$, one-way ANOVA; Figure 3B). (Similarly, treatment with LPS (10 ng/mL; 24 h) as a positive control elicited significant IL-6 increases for dU937, dTHP-1 and dMM6). As expected, the magnitudes of these SP-evoked or LPS-evoked IL-6 responses were greater in macrophage-like cells than in the corresponding undifferentiated monocytic cells). The extent of the IL-6 responses varied between the three cell lines, with two-way ANOVA analysis showing that both SP dose and cell type exhibited significant influences on IL-6 release ($p < 0.001$).

Discrepancies between relatively modest increases in [IL-6]$_{serum}$ levels and severity of COVID-19 symptoms have led to support for specifically evaluating blood-borne levels of the IL-6/sIL-6R 'active complex' (rather than IL-6) as a variable linked to COVID-CSS severity/mortality [4]. Previous studies (see, e.g., Garbers, 2011 [45]) have used a formula based on the dissociation constant ($K_d$) of IL-6 for sIL-6R to calculate samples' concentrations of the IL-6/sIL-6R 'active complex'. Application of this formula to the IL-6 and sIL-6R ELISA data described above is shown in Figure 4: as for sIL-6R (see Figure 2A,B), basal levels of the IL-6/sIL-6R 'active complex' exhibited statistically significant MM6 > THP-1 > U937 patterns ($p < 0.05$, one-way ANOVA). Importantly, this MM6 > THP-1 > U937 pattern, observed for both sIL-6R and for IL-6/sIL-6R 'active complex', indicates that the impact of the rs2228145 genotype on sIL-6R shedding [13,21,22,24,43] results in higher baseline levels of IL-6/sIL-6R 'active complex' (and hence higher potential for trans-signalling) in tissue-culture supernatants from cells with the CC genotype (in this case, MM6) than those with the AA genotype (in this case, U937), with heterozygote cells (in this case, THP-1) showing intermediate levels.

SP treatment also elicited dose-dependent increases in IL-6/sIL-6R 'active complex' levels in all three differentiated macrophage-like cell lines ($p < 0.05$ in all cases; Figure 4A), and in all but one of the monocytic cell lines ($p < 0.05$ for MM6 and U937; the increase seen for THP-1 did not achieve statistical significance; Figure 4B). For both monocytic and macrophage-like cells, the difference between the higher IL-6/sIL-6R 'active complex' levels seen for MM6 versus the corresponding THP-1 and U937 samples was maintained as the SP dose increased. For both datasets, two-way ANOVA analysis indicated that both SP dose and cell-type/rs2228145 genotype exerted significant influences on IL-6/sIL-

6R 'active complex' levels, and hence on samples' potential for IL-6 trans-signalling in response to SP treatment. Importantly, pre-treatment of all three monocytic cell-types with an ACE2 blocking antibody was associated with significant dose-dependent suppression of SP-elicited increases in IL-6/sIL-6R 'active complex' levels ($p < 0.05$, one-way ANOVA; see Figure 4C), supporting our hypothesis that interactions between the SP and the ACE2 receptor are the initial events that trigger these IL-6 trans-signalling responses.

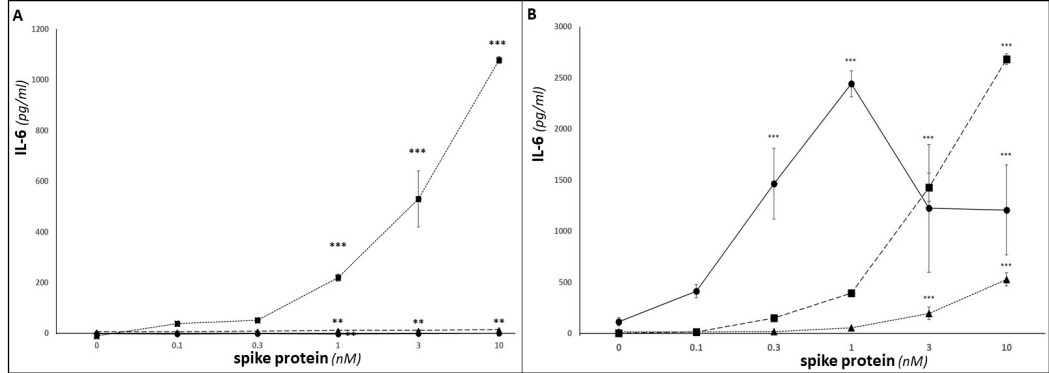

**Figure 3. Determination of IL-6 and sgp130 release from SP-treated U937, THP-1 and MM6.** ELISA duoset assays were conducted as per manufacturers' instructions to determine IL-6 levels in supernatants from U937 (circles; solid line), THP-1 (triangles; dotted line) and MM6 (squares; dashed line) monocytic cultures (**A**), or differentiated dU937 (circles; solid line), dTHP-1 (triangles; dotted line) and dMM6 (squares; dashed line) macrophage cultures (**B**), following treatment for 24 h with 0–10 nM SP. ($n = 6$ in all cases; *** denotes $p < 0.001$; ** denotes $p < 0.01$; asterisks denote comparisons versus the untreated control for the cell line in question).

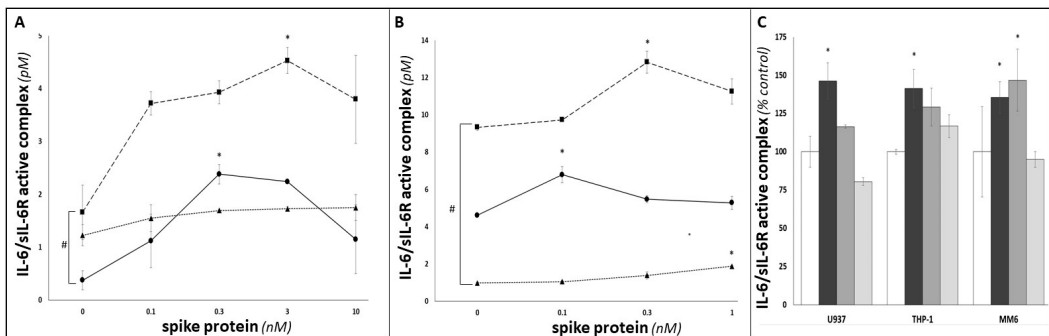

**Figure 4. Investigation of the impact of SARS-CoV-2 SP treatment on IL-6 trans-signalling in U937, THP-1 and MM6 cultures.** Data from ELISA duoset assays were inserted in Garbers et al. (2011)'s formula for calculating concentrations of the 'IL-6/sIL-6R active complex' [30], and hence to investigate the extent of IL-6 trans-signalling in U937 (circles; solid line), THP-1 (triangles; dotted line) and MM6 (squares; dashed line) monocytic cultures (**A**), or differentiated dU937 (circles; solid line), dTHP-1 (triangles; dotted line) and dMM6 (squares; dashed line) macrophage cultures (**B**), following 24 h treatment with SARS-CoV-2 SP. (**C**) shows 'IL-6/sIL-6R active complex' concentrations in supernatants from untreated cultures (white bars), SP-treated cultures (24 h; 1 nM; black bars) and cultures that had been pre-treated (2 h) with ACE2 blocking antibody (2 µg/mL (dark grey bars) or 20 µg/mL (light grey bars)) prior to SP treatment ($n \geq 3$ in all cases; * or # denotes $p < 0.05$; asterisks denote comparisons versus the untreated control for the cell line in question; hash symbols denote comparisons between cell-lines).

Previous studies have identified phosphorylation of pSTAT3-Tyr[705] as a key downstream target of IL-6 trans-signalling [14]. Therefore, we used Western blotting analysis of protein extracts from SP-treated U937, THP-1 and MM6 cultures to assess the extent to which SP treatment triggered STAT3 phosphorylation. Figure 5A shows a representative

Western blot image: while approximately equal intensities were seen for β-actin and total STAT3 in all samples, LPS positive control and MM6 showed markedly stronger p-STAT3-Tyr[705] intensities. (It should be acknowledged that other inflammatory process(es) may also influence STAT3 phosphorylation in monocytes [48] and that therefore the observed difference in this parameter cannot be definitively attributed solely to the rs2228145 genotype's influence on IL-6 trans-signalling. However, it is noteworthy that the respective intensity of STAT3 phosphorylation levels in MM6, THP-1 and U937 (see Figure 5) align with the IL-6/sIL-6R levels (see Figure 4) observed in these cell-lines).

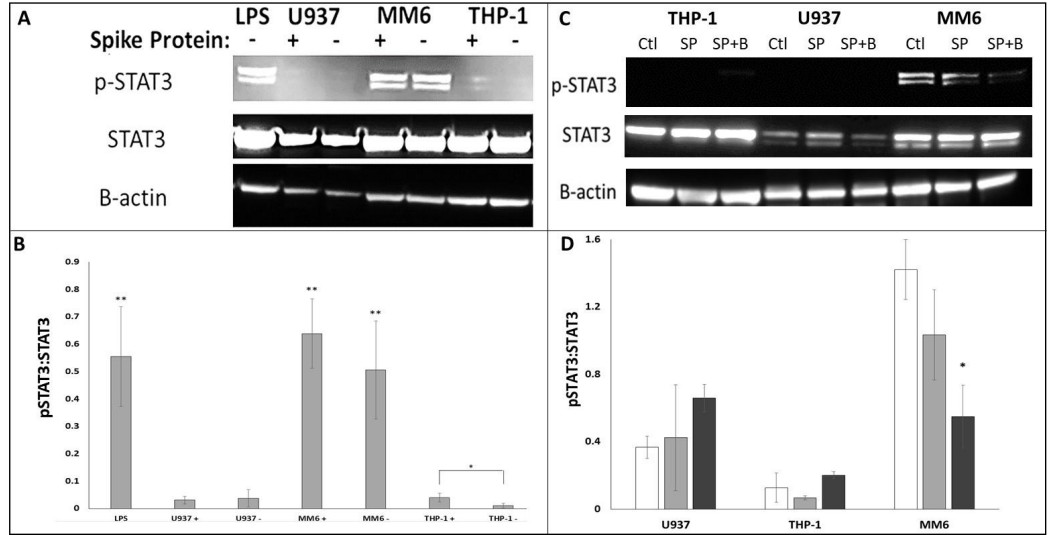

**Figure 5. Investigation of STAT3 Tyr[705] phosphorylation in U937, THP-1 and MM6 cells via Western blot analysis.** Western blot experiments were performed as described in Methods, using antibodies to pSTAT3-Tyr[705], total STAT3 and β-actin. (**A**) Representative image of a Western blot, showing STAT3 Tyr[705] phosphorylation (visible as a doublet of apparent molecular weight approx. 85 KDa; upper row), STAT3 expression (apparent molecular weight approx. 80 KDa; middle row) and β-actin as a loading control (apparent molecular weight approx. 40 KDa; lower row) in U937, THP-1 and MM6 cultures $\pm$ SP treatment (or LPS-treated THP-1 as a positive control). (**B**) Bar graph summarising quantitative band intensity data (obtained using Image J densitometry software; $n \geq 4$ in all cases; asterisks denote comparisons versus untreated THP-1 (negative control); * denotes $p < 0.05$; ** denotes $p < 0.01$). (**C**) Representative image of a Western blot, showing pSTAT3-Tyr[705], STAT3 expression and β-actin (as a loading control) from in U937, THP-1 and MM6 cultures (untreated ('Ctl'), SP-treated ('SP') and cultures that had been pre-treated with ACE2 blocking antibody (20 μg/mL; 2 h) prior to SP treatment ('SP+B')). (**D**) Densitometric summaries of pSTAT3-Tyr[705]: total STAT3 ratios in U937, THP-1 and MM6 cultures (Ctl: white bars; SP: black bars; SP+B: grey bars). ($n \geq 3$ in all cases; asterisks denote comparisons versus the respective SP-treated culture; * denotes $p < 0.05$).

Densitometry was used to convert band intensities into numerical values (summarised in Figure 5B). The summarised data showed significantly increased STAT3-Tyr[705] phosphorylation in LPS positive controls ($p < 0.01$, *t*-test), in either untreated or SP-treated MM6 ($p < 0.01$, one-way ANOVA in both cases), and to a lesser extent in SP-treated THP-1 ($p < 0.05$, one-way ANOVA), while in untreated THP-1 and in U937, no phosphorylation was observed. Thus, it appears that in the U937 cell line, STAT3 was dephosphorylated in the presence or absence of SP (or at least its phosphorylation levels remained below the sensitivity level of our Western blot system), whereas in the THP-1 cell line, STAT3 phosphorylation became activated only when treated with SP (possibly in response to active complexes released from cells within the culture that had come into contact with SP—if so, this would suggest that autocrine and/or 'paracrine' IL-6 trans-signalling responses were occurring throughout SP-treated cultures). Interestingly, MM6 cell cultures appeared to

exhibit constitutive trans-signalling: sufficient IL-6/sIL-6R 'active complex' is produced even in untreated MM6 cultures for STAT3 phosphorylation to be evident in the presence or absence of SP treatment. As shown in Figure 5C,D, STAT3 phosphorylation was decreased via pre-treatment with ACE2 blocking antibody in MM6 cells ($p < 0.05$, one-way ANOVA). While this was not observed for THP-1 or U937, this is likely because the initial response for these two cell lines was at the limit of the sensitivity of our Western blot system, which made visualisation of the blocking of this response impractical. Nevertheless, our MM6 data further support our hypothesis that interactions between the SP and the ACE2 receptor are the initial events that trigger the IL-6 trans-signalling responses under investigation.

Finally, to investigate whether the above effects may be conveyed away from the initial point of SARS-CoV-2 infection, we employed a conditioned media system, in which the responses of A549 lung epithelial cells to treatment with tissue culture supernatants from SP-treated U937, THP-1 or MM6 cultures were determined. In this way, we hoped to mimic endocrine signalling, in which responses are evoked at sites such as the pulmonary tissues where COVID-19 symptoms are experienced (which may be anatomically distant from the site of SP exposure and IL-6/sIL-6R generation) [5]. Production of the pro-inflammatory cytokine IL-8 by A549 cells increased in a significant dose-dependent manner when treated with media from SP-treated MM6 cultures ($p < 0.05$; Figure 6A); however, only trends towards increase that did not achieve statistical significance were seen when A549 cells were treated with media from SP-treated THP-1 or U937 cultures. Two-way ANOVA analysis indicated that the rs2228145 genotype of the monocytes exerted a significant influence on the magnitude of the A549 cells' IL-8 response ($p < 0.001$, two-way ANOVA), with supernatants associated with MM6 samples (i.e., CC genotype) eliciting the strongest responses. We confirmed the role of IL-6 trans-signalling in mediating this IL-8 response, by showing that direct treatment of A549 cells with recombinant SP did not induce a significant response ($p > 0.05$; Figure 6B), whereas when A549 cells were directly treated with recombinant IL-6/sIL-6R complex, the complex directly induced a significant increase in IL-8 release ($p < 0.001$; Figure 6B).

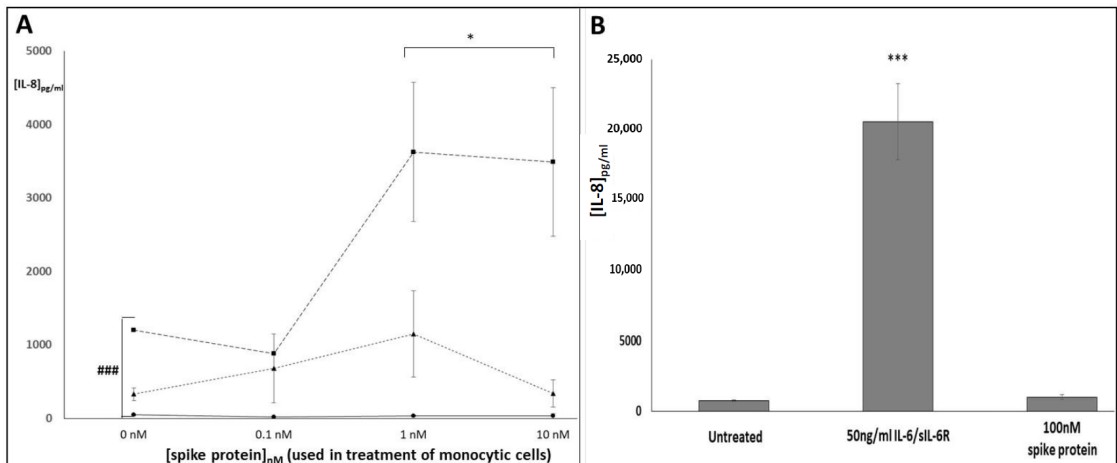

**Figure 6. Conditioned media experiment investigating the impact of tissue-culture supernatants from SARS-CoV-2 SP-treated U937, THP-1 and MM6 cells on release of IL-8 from A549 lung epithelial cells.** ELISA duoset assays were conducted as per manufacturers' instructions to determine: (**A**) Levels of IL-8 release from A549 cultures, following treatment with supernatants from U937 (circles; solid line), THP-1 (triangles; dotted line) and MM6 (squares; dashed line) monocytic cultures that had been exposed to SARS-CoV-2 SP (0–10 nM; 24 h). Levels of monocyte-derived IL-8 (calculated separately) were subtracted from the total IL-8 levels seen in supernatants from the A549 cultures, as a post-hoc method of calculating the levels of A549-derived IL-8. (**B**) IL-8 levels in supernatants from A549 cultures treated directly with media ('untreated'), 10 nM SP or 50 ng/mL IL-6/sIL-6R ($n = 3$ in all cases; *** or ### denotes $p < 0.001$; * denotes $p < 0.05$; asterisks denote comparisons versus the untreated control for the cell line in question; hash symbols denote comparisons between cell-lines).

## 4. Discussion

As stated earlier, because U937, THP-1 and MM6 span the range of rs2228145 genotypes that occur in vivo, we employed these three cell lines as an in vitro model system to gain insights regarding the influence of rs2228145 on the responses of monocyte-macrophages from patients infected with SARS-CoV-2. Our data indicate that rs2228145 does indeed influence responses to SP treatment: an MM6 > THP-1 > U937 pattern (which equates to CC > AC > AA, in terms of rs2228145 genotype) was consistently observed across our experiments. rs2228145-influenced effects included sIL-6R shedding, formation of the IL-6/sIL-6R 'active complex' and downstream effects of IL-6/IL-6R trans-signalling such as STAT3 phosphorylation and triggering of IL-8 release from A549 cells. Additionally, the demonstration that all three cell lines express comparable levels of ACE2 and the suppression in all three cell types of the above effects by ACE2-blocking antibodies support the hypotheses that: (i) these effects are downstream sequelae of SP/ACE2 interactions and (ii) variations in the magnitude of these effects in the different cell lines may be tentatively attributed to the different rs2228145 genotypes of MM6, THP-1 and U937.

However, such an interpretation must be subject to caveats. Firstly, as an in vitro study utilising immortalised cell lines, this study has limitations. Neither MM6, THP-1 nor U937 are fully representative of primary monocyte-macrophages [49–51], and so their responses may not accurately reflect the in vivo responses of primary monocyte-macrophages to SARS-CoV-2 infection. Furthermore, use of recombinant SARS-CoV-2 spike protein (SP) as an in vitro mimic of SARS-CoV-2 infection may not fully reproduce the intricate and multifaceted responses of cells to viral infection. (However, it should be acknowledged that previous studies have reported comparable increases in monocyte-macrophage IL-6 release, ADAM-17 expression, sIL-6R shedding and downstream cytokine release in other cell-types (such as endothelial cells and Th17 lymphocytes) in response to viral infection or SP treatment [14,35], which supports the view that, in this case, SP treatment may be considered an adequate in vitro surrogate for viral infection.)

Secondly, in addition to ACE2, SARS-CoV-2 may use alternative host receptors/co-receptors for cell attachment and entry [52]. Interactions with such alternative receptors may exert additional influences on the monocyte-macrophage response to SP; however, such additional influences have not been investigated in this study.

Thirdly, the interpretation given in the previous paragraph could be viewed as depending on an assumption that MM6, THP-1 and U937 cells are identical except for their rs2228145 genotype and that, therefore, differences between their responses must be attributable solely to their different rs2228145 genotypes. However, this is not the case: these three immortalised cell lines were originally derived from different individuals with distinct malignant backgrounds, and they are known to possess different phenotypic and functional characteristics [36–38]. Therefore, it is likely that some of the observed differences between MM6, THP-1 and U937 stem from background/immortalisation-related impacts, rather than from their different rs2228145 genotypes.

Despite these caveats, we would contend, given the clear role for IL-6R in the responses that exhibit an MM6 > THP-1 > U937 pattern, that it is reasonable to attribute several of the effects observed in this study to the influence of rs2228145. As described in the Introduction section, rs2228145 involves an A to C change at position 1073 of the IL-6R gene sequence, leading to an $Asp^{358} \rightarrow Ala^{358}$ change in the IL-6R protein, which enhances cleavage of mbIL-6R at the adjacent cleavage site ($Val^{356}$), and hence increases sIL-6R shedding [13,21,22,43]. It should be noted that van Dongen et al. have calculated that over 50% of the variability in sIL-6R levels is explained by rs2228145′s influence [22]. This influence is reflected in the statistically significant MM6 > THP-1 > U937 pattern of basal sIL-6R levels (and levels of the IL-6/sIL-6R 'active complex') seen in the tissue-culture supernatants from the cell lines (Figures 2 and 4).

Importantly, this pattern is maintained in SP-treated cells; this suggests that, in addition to sIL-6R shedding occurring at higher baseline levels, SARS-CoV-2 infection-triggered sIL-6R shedding would occur at higher levels in patients with the rs2228145 CC genotype.

Therefore, taken together with reports that COVID-19 patients suffering from COVID-CSs experienced much larger sIL-6R increases (up to >125 ng/mL) than those with milder symptoms [16], our findings suggest that patients with CC genotypes might be expected, on average, to experience more severe symptoms than would bearers of AA or AC genotypes.

As described earlier, it is increasingly becoming clear that the IL-6 signalling system is involved not only in systemic aspects of COVID-19 such as COVID-CSs but also in the neuropsychiatric aspects of long-COVID [10,11,31–33]. While in vivo levels of IL-6 are similar in plasma and CSF [30], CSF levels of sIL-6R (~0.5–2 ng/mL, versus ~20–40 ng/mL in plasma) and sgp130 (~30 ng/mL, versus ~300 ng/mL in plasma) have consistently been reported as significantly lower than in the blood [53,54]). While, according to Garbers (2015), IL-6, sIL-6R and sgp130 form a blood-borne 'IL-6 buffer complex' (with 2:2:2 stoichiometry [13]) whose buffering capacity is determined by the sIL-6R concentration [46], this depends on circulating sgp130 levels remaining in excess at all times. Therefore, although in some contexts rs2228145 genotypes linked to high sIL-6R levels are associated with enhanced buffering and hence can be protective [46,55], the lower $[sgp130]_{CSF}$ levels stated above suggest that this may not be the case in the CNS.

In our study, basal IL-6 levels were comparable to those previously reported for plasma and cerebrospinal fluid (CSF) (i.e., $\leq$10 pg/mL; see Figure 3), while sIL-6R and sgp130 were present in tissue-culture supernatants at similar levels to those previously reported for the CSF: namely, ~0.1–1 ng/mL for sIL-6R and ~12–18 ng/mL for sgp130 (see Figure 2A–C; these low levels may be due to only one cell type being present in each cell-culture sample (i.e., the absence of non-monocyte-macrophage sources of sIL-6R or sgp130 must be considered), and possibly removal of sIL-6R and sgp130 during sub-passaging). Moreover, we observed stronger trans-signalling responses in CC-bearing MM6 cells than in cells bearing other genotypes (see Figures 4–6). This suggests that in our in vitro system, the IL-6/sIL-6R active complex may not be fully buffered by sgp130, with the greater levels of unbuffered active complex in MM6 cultures directly triggering trans-signalling responses to a greater extent than in THP-1 or U937 (where levels of un-buffered active complex will be lower). Importantly, the similarity of our sgp130 values to those seen in CSF suggest that our in vitro model could be applied to gain insights regarding the impact of the rs2228145 genotype on in vivo inflammatory IL-6 signalling in the CSF. It is therefore interesting to note that recent studies have identified the rs2228145 CC genotype as the at-risk genotype in contexts including Alzheimer's disease [29], impaired cognitive performance [29] and inability to tolerate exercise training interventions (likely linked to differences in perception of fatigue/susceptibility to stress) [43,56,57], while direct correlations have been reported between sIL-6R levels and aspects of mental wellbeing such as chronic fatigue syndrome, stress and PTSD [27,28,57–59].

In a COVID-19 context, therefore, we hypothesise that the rs2228145 CC genotype may place patients at greater risk of developing the neuropsychiatric symptoms of long-COVID. Clearly, given the in vitro nature of this study, future research is needed to test this hypothesis by performing clinical studies to determine patients' rs2228145 genotypes and investigate whether such an association exists. Other future investigations could determine whether the rs2228145 genotype of non-monocyte-macrophage cell types, including CNS cell types, influences their responses to SARS-CoV-2 infection (or treatment with SARS-CoV-2 SP as an in vitro surrogate for infection).

## 5. Conclusions

In conclusion, the current study provides a model for how rs2228145 influences COVID-19 pathology, perhaps particularly the likelihood of developing the neuropsychiatric symptoms of long-COVID, and tentatively identifies the rs2228145 CC genotype as the at-risk genotype in this context. This study points towards the possible future use of the rs2228145 genotype as a biomarker predictive of severe COVID-19 and/or long-COVID risk. If so, the exciting prospect emerges that the rs2228145 genotype (which can be determined via a simple mouthwash-DNA extraction-PCR/RFLP procedure [43]) may be of use in

identifying at-risk patients, facilitating rapid signposting towards treatment options that can combat the relevant symptoms and bringing advantages regarding the management and treatment of COVID-19 and/or long-COVID.

**Supplementary Materials:** The following supporting information can be downloaded at: https://www.mdpi.com/article/10.3390/covid3100106/s1, Figure S1: Determination of ACE2 and ADAM-17 expression in dU937, dTHP-1 and dMM6 via Flow Cytometry. Representative 'cell count [i.e., '*events*'] v. fluorescence' histograms are presented as 'cells only' (grey trace), 'isotype control' (black trace) and 'ACE2 expression' (red trace) for dU937 (**A**), dTHP-1 (**B**), dMM6P-1 (**C**) and as 'cells only' (grey trace), '$2^0$ antibody only' (black trace) and 'ADAM-17 expression' (red trace) for dU937 (**D**), dTHP-1 (**E**) and dMM6 (**F**). No significant differences in ACE2 or ADAM-17 surface expression between the three cell lines were detected ($n = 3$; $p > 0.05$, Kruskal–Wallis); Figure S2: Investigation of NFkB p65 Ser$^{276}$ phosphorylation in U937, THP-1 and MM6 cells via Western blot analysis and Bioinformatics analysis of IL-6 and ADAM-17 as potential NFkB target genes. Western blot experiments were performed as described in Methods, using antibodies to NFkB p65 pSer$^{276}$, NFkB p65 and β-actin. (**A**) Representative image of a Western blot, showing NFkB p65 Ser$^{276}$ phosphorylation (apparent molecular weight approx. 80KDa; upper row), NFkB expression (apparent molecular weight approx. 65KDa; middle row) and β-actin as a loading control (apparent molecular weight approx. 45KDa; lower row). (**B**) Bar graph summarising quantitative band-intensity data (obtained using Image J densitometry software; $n = 4$; *denotes $p < 0.05$ for THP-1 untreated vs. LPS). Inset: Sequence analyses to identify NFkB-RE motifs (5′-GGGATTTTCC-3′) in promotor regions of IL-6 and ADAM-17 genes were performed using DNASTAR software (Lasergene, version 7; DNASTAR Inc., WI, USA). NFkB-RE element sequences were found to be present at position −136 of the IL-6 gene (also at position −4285), and at position −1073 of the ADAM-17 gene. All sequences were obtained from National Center for Biotechnology Information databases (http://www.ncbi.nlm.nih.gov/; accessed 22 July 2023).

**Author Contributions:** Conceptualization, R.A., L.B. and R.W.; methodology, R.A., L.B., S.P., R.R. and R.W.; validation, R.A., L.B., S.P., R.R. and R.W.; formal analysis, S.S., K.R. and R.W.; investigation, S.S., K.R., S.P., R.R. and R.W.; resources, R.A., L.B., S.P. and R.W.; data curation, R.W.; writing—original draft preparation, R.W.; writing—review and editing, S.S., R.A., L.B., K.R., S.P., R.R. and R.W.; visualization, S.S., K.R. and R.W.; supervision, R.A., L.B. and R.W.; project administration, R.A., L.B. and R.W. All authors have read and agreed to the published version of the manuscript.

**Funding:** This research received no external funding.

**Data Availability Statement:** All data was obtained within Cardiff Metropolitan University laboratories. Source data is available on request.

**Acknowledgments:** The excellent supporting work of Gareth Walters, Elizabeth Decoine, Joshua Davies, Sam Hooper, Paul Jones and Sean Duggan is gratefully acknowledged.

**Conflicts of Interest:** The authors declare no conflict of interest.

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
