# Peer review of "The rs2228145 Variant of the Interleukin-6 Receptor (IL-6R) Gene Impacts on In Vitro Cellular Responses to SARS-CoV-2 VOC B1.1.7 Recombinant Spike Protein"

_covid, doi:10.3390/covid3100106_

Round 1

Author Response

We have revised our manuscript in accordance with Reviewer 1’s comments (see attached amended manuscript); we have highlighted our revisions in red text and by using ‘Track changes’. In the attached 'Responses to Reviewers' document, we will respond to these comments on a comment-by-comment basis.

Reviewer 2 Report

The authors presented an in-vitro model system based on monocyte-macrophages to investigate the impact of a clinically-relevant polymorphism, rs2228145, on the body's inflammatory response to SARS-CoV-2. This study holds significant value as it provides an insight into the intricate relationship between inflammatory responses and the development of COVID-19 disease.

Here are some suggestions and questions:

1) Cytokine storms in COVID-19 can have devastating consequences, leading to severe complications such as acute respiratory distress syndrome (ARDS), multi-organ failure, and even death. Given their significance in the development of COVID-19, it is crucial for authors to include more information on cytokine storms in the introduction part of their article.  (line 65)

2) Line 131: It seems the statement on AA, AC and CC genotypes of the cells are not consistent with the characterization results in Fig. 1

3) The authors refrained from conducting infection experiments on the three cell lines. Considering this, I proposed that they discuss the limitations of using SP treatment in comparison to infection assays. It is important to acknowledge that IL-6 related responses might be more intricate and multifaceted when the cells are infected with the virus, as opposed to when they undergo SP treatment alone.

4) MM6 cells exhibited strong IL-6 responses before differentiation. Additionally, it seems that the IL-6 response increased dramatically of U937 after it had differentiated into macrophage.  Can this point be explained by the genotype differences? I proposed that the authors discuss this point. (Fig. 3)

5)  The authors reported significant differences in the levels of STAT3-Tyr705 phosphorylation between MM6 cells with the CC genotype and the other two cell lines (Fig. 5). However, it may be beneficial for the authors to provide further discussion on this point. The evidence presented in the study is not sufficient to conclusively attribute this difference solely to the CC genotype and higher levels of IL-6/sIL-6R.

Minor points:

1) The size of the markers in the line chart are too small. It is difficult to read.

Author Response

(The authors gave the same response as above.)
